# A Study of Projection-Based Attentive Spatial–Temporal Map for Remote Photoplethysmography Measurement

**DOI:** 10.3390/bioengineering9110638

**Published:** 2022-11-02

**Authors:** Dae-Yeol Kim, Soo-Young Cho, Kwangkee Lee, Chae-Bong Sohn

**Affiliations:** 1AI Research Team, Tvstorm, Seoul 13875, Korea; 2Department of Electronics and Communications Engineering, Kwangwoon University, Seoul 01897, Korea; 3Department of Information Contents, Kwangwoon University, Seoul 01897, Korea

**Keywords:** remote photoplethysmography, CVD, axis projection

## Abstract

The photoplethysmography (PPG) signal contains various information that is related to CVD (cardiovascular disease). The remote PPG (rPPG) is a method that can measure a PPG signal using a face image taken with a camera, without a PPG device. Deep learning-based rPPG methods can be classified into three main categories. First, there is a 3D CNN approach that uses a facial image video as input, which focuses on the spatio-temporal changes in the facial video. The second approach is a method that uses a spatio-temporal map (STMap), and the video image is pre-processed using the point where it is easier to analyze changes in blood flow in time order. The last approach uses a preprocessing model with a dichromatic reflection model. This study proposed the concept of an axis projection network (APNET) that complements the drawbacks, in which the 3D CNN method requires significant memory; the STMap method requires a preprocessing method; and the dyschromatic reflection model (DRM) method does not learn long-term temporal characteristics. We also showed that the proposed APNET effectively reduced the network memory size, and that the low-frequency signal was observed in the inferred PPG signal, suggesting that it can provide meaningful results to the study when developing the rPPG algorithm.

## 1. Introduction

A PPG (photoplethysmogram) is measured by analyzing the physical characteristics of hemoglobin in blood that are generated by light transmission and reflection. Using PPG information, various vital signals such as pulse, oxygen saturation, blood pressure, and respiration rate can be acquired. Furthermore, stroke symptoms are detected through frequency analysis, renin-angiotensin-based variables, sympathetic and parasympathetic nervous system components; from these, respiratory arrhythmias can be detected. The disease information included according to the frequency band of the PPG is shown in Table 1 [1]. PPG measurement is divided into an oximeter method, which involves a contact-type device that uses the tip of a finger as the measurement area, and the rPPG method, which observes changes in blood flow in the face via video data. Between the two methods, rPPG is emerging as a new method of measuring PPGs, since it can be easily measured with a smartphone camera, and has the strength of being a non-contact measurement method [2].

Heart diseases such as myocardial infarction and heart failure can be detected by continuously monitoring vital signs. In addition, according to the early warning score (EWS) in Table 2, which is used in medical services to determine the disease level of a patient at an early stage, deterioration of health and cardiac arrest can be detected using vital signs [3]. rPPG is a video-based measurement method that measures changes in blood flow by analyzing the amount of light-diffused reflection information that is generated by changes in blood flow in the face. This approach benchmarks the principle of PPG sensors, but overcomes the problem in which a PPG may not be suitable for continuous monitoring, as the measuring device must always be used in a contact manner. Since the rPPG can be measured using a camera, it can effectively be used in any space, as a result of the spread of mobile devices and IOT devices that have cameras.

With the development of deep learning, various rPPG models are being studied. Among the rPPG models, representative methods that are used as baselines include “PhysNet”, “RhythmNet”, and “Deephys”; each has a unique approach [4,5,6]. “PhysNet” analyzes the spatio-temporal characteristics of video pixel changes with a 3D convolutional network; “RhythmNet” learns the time series characteristics of PPGs by introducing a preprocessing process that converts video into STMap; finally, “DeepPhys” uses a differential image learning method that removes specular reflection information, and learns the amount of change in the PPG waveform based on the DRM model. As far as the problems of each method are concerned, “PhysNet” uses a lot of memory because it uses 3D convolution; “RhythmNet” involves a large proportion of preprocessing that occurs, since it uses STMap; finally, since “DeepPhys” infers two image groups, it cannot learn long-term time series features, and consequently requires post-processing.

In this study, we propose the concept of an axis projection network that complements the various problems of representative models. The suggestions are as follows:Axial Projection:

Proposal of a new time-series feature analysis method that uses axial projection. By projecting the video onto various axes, the information obtained from the data in each direction is acquired.

End-to-end rPPG inference model:

Analyze facial features and perform spatio-temporal feature analysis, with a performance that is similar to STMap at the network level.

Memory saving model:

Minimize the parameters of the deep learning model using only the minimum number of 3D convolutions.

## 2. Related Research

Various rPPG methods are being studied, with the goal of achieving similar performance to contact PPG (cPPG), i.e., the results of an oximeter [7,8]. Although mathematical models, such as CHROM, that use the existing color difference information and POS with light reflection models showed excellent performance, they showed poor performance in various bad environments, depending on motion artifacts by human movement, and light distortion. In order to overcome these problems, research is underway to create an rPPG model that uses deep learning [9,10].

### 2.1. Deep Learning-Based rPPG Methods

Various deep learning methods are being studied for rPPG tasks, which can be divided into three main methods, according to the data input type.

The first is a method that obtains differential information from an image or video as a model’s input. “HR Track” analyzes the difference image spatially, using 3D CNN as video information, and “DeepPhys” is a 2D CNN model that uses a single difference image as input. In particular, the “DeepPhys” is modeled on the basis of the DRM, and it consists of a motion model that extracts features from the degree of movement between two frames from a face video, and an appearance model that extracts facial features through normalization values between two frames [6,11]. Derived from this method, a “high-order” study was conducted [12]. This suggested a study to obtain the first derivative PPG (VPG), and the second derivative PPG (APG), which can provide additional information, such as blood pressure estimation and heart rate fibrillation, from facial images [13].

The second method involves using the video as an input without preprocessing. “PhysNet” devised a 3D CNN-based network to learn the spatio-temporal features of the face, and to measure the PPG signal. It uses the neg-Pearson loss, and through this, improved experimental results are derived. Afterwards, performance is improved through heart rate inference, using the spatial temporal efficient network (STVEN), using the modified 3D CNN [4,14].

The last method uses the STMap format, which preprocesses the video in 2D imagery, as an input. “RhythmNet” uses a preprocessing method with STMap. The PPG is measured through 2D CNN by converting video into STMap, showing spatio-temporal features [5].

### 2.2. Transformer Network in rPPG Tasks

The transformer module was first proposed in the field of NLP, and showed good results. Recently, Vision Transformer (ViT) has been introduced and widely used as a method for image learning in vision tasks. An image patch-based inference method was proposed for image classification, and various variants of ViT were proposed to resolve the characteristic that ViT lacks inductive bias [15,16,17,18]. Various ViTs showed better performance than existing CNNs in image analyses, and ViTs for motion recognition, detection, and super resolution were recently introduced. In the rPPG task, there are related studies that used ViT for rPPG inference, such as “Transrppg”, “EfficentPhys”, and “PhysFormer” [19,20,21]. “Transrppg” extracts the rPPG from the space that is preprocessed through ViT for face attack detection in rPPG. “EfficientPhys” extracts spatial–temporal features using the Swin layer, and “Physformer” measures tube Tokenizer-based long-distance spatial–temporal rPPGs in facial video, and showed good results.

## 3. Materials and Methods

In this section, the methodology of this study is discussed. Three drawbacks in representative deep learning methods for rPPG tasks are considered in this study. The datasets are preprocessed before model training. All of the video data are separated into the proposed model’s T window. The preprocessed data are then used to train APNET. Section 3.1 describes the structure of the APNET, and Section 3.2 describes the dataset used in this paper and the preprocessing process. Section 3.3 describes the proposed loss function, and finally, Section 3.4 describes the metric for evaluating the proposed APNET.

### 3.1. APNET: Axis Projection Network Architecture

In this study, we proposed the concept of APNET, which does not require any other preprocessing, and analyzes video features from various directions using axial projection.

Figure 1 shows APNET architecture. The APNET is composed of axis feature extractor, feature mixer, and PPG decoder block, which are feature extractors for each axis that have the same shape. The transformer learns global features, but lacks inductive bias and tends to overfit the training set. CNN learns inductive bias, but does not learn global features. We used MaxViT, because it has the characteristic of learning local features at the same time as learning global features [22]. Specifically, axis feature extractor is designed to extract features that are specialized in one axis. Given an RGB face video input, each feature is extracted through $X_{Video}\in R^{B\times 3\times T\times H\times W}$. The output of the feature extractor is combined through the feature mixer, and at this time, it is designed to create the optimal feature through various calculations. The optimal feature is transferred to a one-dimensional PPG block for generating the PPG. PPG block functions to decode the processed feature into a PPG of *T* length.

#### 3.1.1. Axis Feature Extractor: Axis Projection-Based Time-Series Feature Extractor Module

Axis feature extractor is designed to have the same model structure, with only different input types for each axis. Stream is composed of a 2D feature extraction block, *ST* (spatial–temporal) translator, and *ST* feature extraction block. The input of the model proceeds in the form of a three-dimensional video arrangement, but each input is reconstructed into a two-dimensional form, and used for the input of the 2D feature extraction block.
(1)$$X_{Video}\in R^{B\times 3\times T\times H\times W}\to X_{Feature\;Extractor}^{n}\in R^{\left(B\times D_{1}^{n}\right)\times 3\times D_{2}^{n}\times D_{3}^{n}}\;,\;n\;\in \left\{\;T,\;H,\;W\right\}\;D^{M}\in \left\{\;T,\;H,\;W\right\},\;\;D^{H}\in \left\{\;H,T,\;W\right\},\;\;D^{W}\in \left\{\;W,\;T,\;W\right\}$$

The 2D feature block consists of two Conv2D operations using kernel size (3 × 3) and stride size 2.
(2)$$E_{2D\;FeatureBlock}\left(X_{Feature\;Extractor}^{n}\right)=Conv2D\left(Conv2D\left(X_{Feature\;Extractor}^{n}\right)\right)\;=X_{2D\;FeatureBlock}^{n}\in R^{\left(B\times D_{1}^{n}\right)\times 3\times D_{2}^{n}/4\times D_{3}^{n}/4}$$

*ST* translator functions to change the 2D image into STMap format.
(3)$$E_{ST\;Translator}\left(X_{2D\;FeatureBlock}^{n}\right)=X_{ST\;Translator}^{n}\in R^{B\times 3\times D_{1}^{n}\times \left(D_{2}^{n}/4\times D_{3}^{n}/4\right)}$$

The *ST* feature extraction block is composed of kernel (1, *T*), MaxViT layer with dilation (1, *T*), spatial attention module, and 3D translator cascade. The *ST* Feature extraction block provides the effect of learning spatio-temporal features using STMap form data.
(4)$$E_{\mathrm{ST}\;\mathrm{FeatureBlock}}^{n}\left(X_{\mathrm{ST}\;\mathrm{Translator}}^{n}\right)=\;\mathrm{SpatialAttention}\left(\mathrm{MaxVit}\left(X_{\mathrm{ST}\;\mathrm{Transformer}}\right)\right)\;$$

#### 3.1.2. Feature Mixer: Spatio-Temporal Feature Combination Module for Generating Attentive Feature

Each of the three features that is extracted from axis feature extractor has specific features for the *T*-axis, *W*-axis, and the *H*-axis. It is difficult to expect high accuracy when learning a PPG using only each feature, but high accuracy can be expected by combining the facial characteristics extracted from the time axis, the PPG characteristics of the *W*-axis, and the PPG characteristics of the *H*-axis. Interpolation is required, since the results extracted from the stream are processed according to the input size of each axis. After interpolation, the processed feature map is processed, which is an attentive 3D feature that includes PPG information.

Figure 2 is a type of feature mixer. The attentive feature is divided into 10 cases, according to the processing method. The purpose of the feature mixer is to combine the characteristics of each axis, according to the shape of the feature of the time axis. The detailed formula is expressed in Table 3.

#### 3.1.3. PPG Decoder: Generate PPG Signal from Attentive Feature

The attentive feature requires Conv3D operation to extract time series information in three-dimensional form. However, it was changed to STMap, and then Conv2D was applied to achieve a similar performance to Conv3D. In order to learn the features of all of the areas, MaxViT was selected and used, and the last feature to extract the PPG was generated via spatial averaging. The result was obtained through the residual operation of Conv1D.

#### 3.1.4. APNet Configuration

The APNET configurations that were evaluated in this paper are outlined in Table 4, one per column. In the following, we mention each layer’s output shape. All of the configurations followed the generic design that was presented in Section 3.1, and differ only in the feature mixer.

### 3.2. Dataset Description and Preprocessing Method

#### 3.2.1. Dataset Description

Two datasets were used in this study: UBFC and V4V.

UBFC [23]:

The V4V dataset provided data that consisted of 42 participants. The ground truth was provided with PPG information, heart rate. The participants looked directly at a camera installed at a distance of 1 m that recorded video.

V4V [24]:

The V4V dataset provided training data that consisted of 82 female participants and 58 male participants. Each participant performed 10 tasks, including smiling and squinting. The ground truth was provided with BP information, heart rate, and respiration rate, sampled at 1000 kHz. In the actual challenge, the model trained with the training data was evaluated as a test set, but the dataset was composed by dividing the training set into a ratio of 80:20, since the dataset did not contain label information for the test set.

#### 3.2.2. Preprocessing Method

All of the datasets consisted of video data and label data. In order to use video data for APNET training and inference, data resizing and slicing were required in accordance to the input shape of the network configuration. According to the APNET configuration in Table 3, the APNET input type was (B, C, T, H, W), and it was necessary to cut it into T frames. In APNET, faces were detected using the HOG algorithm in every frame [25]. Moreover, since the size of the completely cropped face did not match the input shape of the network, it was resized to (128, 128). Labels also required slicing, in order to fit the output size of the network configuration.

### 3.3. Loss Function

The PPG signal contained various physical information. In order to simply infer the heart rate, information extraction was possible only with neg-Pearson.
(5)$$rPPG^{inf}=APNet\left(X_{video}\right)\;A_{rPPG^{gt}},\;P_{rPPG^{gt}}=FFT\left(rPPG^{gt}\right),\;A_{rPPG^{inf}},\;P_{rPPG^{inf}}=FFT\left(rPPG^{inf}\right)\;L_{fft}=║A_{rPPG^{gt}}-A_{rPPG^{inf}}{║}_{1}+║P_{rPPG^{gt}}-P_{rPPG^{inf}}{║}_{1}\;$$
where $rPPG^{gt}$ are the ground truth label signals, and $rPPG^{inf}$ is the inference value of APNET. A and P refer to the amplitude and phase of the rPPG signal. FFT denotes a fast Fourier transform. The FFT loss has the effect of learning a fine waveform by comparing the size of each frequency band [26].
(6)$$L_{neg-pearson}=1-\frac{T\sum_{1}^{T}rPPG^{gt}\times rPPG^{inf}-\sum_{1}^{T}rPPG^{gt}\sum_{1}^{T}rPPG^{inf}}{\sqrt{\left(T\sum_{1}^{T}rPPG^{gt}{}^{2}-{\left(\sum_{1}^{T}rPPG^{gt}\right)}^{2}\right)\left(T\sum_{1}^{T}rPPG^{inf}{}^{2}-{\left(\sum_{1}^{T}rPPG^{inf}\right)}^{2}\right)}}\;\;$$
where T is the length of the signal. The neg-Pearson loss has the effect of learning the trend similarity and peak position. In previous studies, neg-Pearson was used to target the heart rate measurement using the trend and peak of the PPG. In this study, neg-Pearson + FFT loss was used as the loss, paying attention to the fact that the peak and notch of a PPG are needed to extract various information.

Figure 3 shows the waveform of a PPG. There are three important points in a PPG: systolic peak, diastolic peak, and notch. Notch is an important point for inferring blood pressure. In the frequency domain, notches are represented by low frequencies [27].

### 3.4. Assessment Metric of Proposed Method

#### 3.4.1. Quantitative Assessment Methods

Pearson correlation coefficient (PCC; R): PCC is a method of interpreting the linear relationship between two given signals. The closer the PCC result is to 1, the more positive the linear relationship is.HR-mean average error (MAE): HR-MAE is used to verify the accuracy of HR reprocessed with rPPG results.


(7)
$$HR-MAE=\frac{1}{N}\underset{}{\sum }\left|HR_{inf}-HR_{target}\;\;\right|\;\;$$


HR-root mean square error: This is used to see the standard mean error of the HR.


(8)
$$HR-RMSE=\frac{1}{N}\sqrt{\underset{}{\sum }{\left(HR_{inf}-HR_{target}\;\right)}^{2}}\;\;$$


Network memory: The larger the network memory required, the more unusable it is in a low-spec environment. The smaller the memory, the better the performance of other evaluation indicators.

#### 3.4.2. Subjective Assessment Methods

Roi assessment: The importance of each pixel of the input image is determined using a backpropagation-based method (BBM) [28].

Figure 4 is an image of a PPG extraction accuracy evaluation, according to face area. According to [29], upper medial forehead, lower medial forehead, glabella, right malar, and left malar pixel data are important.

## 4. Results and Discussion

This section contains the assessment results of the experiments, and the results are compared with the published representative algorithms.

### 4.1. Quantitative Assessment Results

Using the three-evaluation metrics mentioned in Section 4.2, we evaluated ten feature mixer types and two loss functions with UBFC and V4V. For fair evaluation, each person’s inference results were concatenated into one signal, and then model evaluation was performed.

Table 5 shows the evaluation results for 10 functional mixers using the V4V dataset. HR-MAE and HR-RMSE were evaluated for heart rates per second. Learning proceeded smoothly, both when only neg-Pearson loss was used and when neg-Pearson and FFT loss were combined and used as a loss function. The combination of feature mixer (1) and neg-Pearson loss showed the highest correlation, and the combination of feature mixer (6) and neg-Pearson loss + FFT loss showed good performance in HR-MAE and HR-RMSE.

Figure 5 shows loss graphs of APNET. (a) shows that when only the neg-Pearson loss function is used, WT + T has the largest loss, and when only W and H are used, no training is performed. (b) uses neg-Pearson loss + FFT loss. The loss of T(W@H) +T is the lowest, and learning is not performed on W, H.

Table 6 shows the evaluation results for 10 functional mixers using the UBFC dataset. HR-MAE and HR-RMSE were evaluated for heart beats per second. Learning proceeded smoothly, both when only neg-Pearson loss was used, and when neg-Pearson and FFT loss were combined and used as a loss function. The combination of feature mixer (4) and neg-Pearson loss showed the highest correlation, and the combination of feature mixer (3) and neg-Pearson loss + FFT loss showed good performance in HR-MAE and HR-RMSE.

Ten feature mixers were trained with neg-Pearson, neg-Pearson + FFT loss with two datasets, UBFC and V4V, and HR-MAE, HR-RMSE, and rPPG correlation evaluations were performed for each. Overall, learning did not proceed stably with feature mixers 9–10, and learning progressed stably with 1–8. All of the feature mixers in 1–8 showed excellent performance. Based on the results of Table 5 and Table 6, when trained with neg-Pearson loss + FFT loss, the results had a correlation value similar to that of neg-Pearson loss; however, HR-MSE showed that HR-RMSE performance improved.

Table 7 shows the quantitative assessment of a representative rPPG algorithm. The proposed APNET showed the best HR-MAE and HR-RMSE values in both datasets, the correlation of PhysNet showed the best performance in V4V, and the correlation of APNET showed the best performance in UBFC.

Table 8 shows the model size of a representative rPPG algorithm. Compared to Deepphys, memory decreased by about 4 MB, and compared to PhysNet, memory decreased by 1.9 MB.

### 4.2. Subjective Assessment Results

LRP (layer-wise relevance propagation), one of the representative BBMs, decomposes the output of DNN to obtain relevance scores for each feature [30]. In CNN, validity is propagated between layers, and it is a method that can indicate the importance of each pixel by calculating the degree of error through backpropagation.

Figure 6 is the result of LRP. The closer the pixel value is to red, the higher the importance value of the deep learning network is evaluated. As a result of LRP, it can be seen that the forehead, glabella and malar regions were selected as the regions of interest, from among the regions that showed good performance in extracting the five types of PPGs mentioned in 4.2.2. It can be seen that the nose, which has a negative effect on learning, was evaluated as a region of interest.

Figure 7 is a graph of the inference result. As a result of the graph, (a) shows a tendency to learn peaks, and (b) shows a tendency to learn low frequency signals other than peaks.

## 5. Conclusions

In summary, we proposed in this study the following:The concept of axis projection that can replace the existing spatio-temporal analysis method.APNET axis projection network and evaluation of 10 feature mixers.A new loss function so that the low-frequency signal of a PPG can be learned, and the importance of the input pixel can be analyzed using the BBM method.

In this study, the concept of axis projection, a new spatio-temporal analysis method, was proposed. Axis projection has the advantage of using less memory, by converting a 3D shape into a 2D shape. In addition, APNET that used axis projection was proposed, and a comparative evaluation was performed through 10 feature mixers to obtain good performance in APNET. Blood flow in the face of a person has a high correlation with respect to the nose, and tends to flow from arteries to capillaries. Therefore, it is possible to extract physiological characteristic information from each block divided into H, W, and T, and excellent results can be obtained if each physiological characteristic information is well extracted.

We also proposed a new loss function, in which the low frequencies of a PPG can be learned. Notch is an important biomarker that can detect LVET (left ventricular ejection time) and blood pressure [31]. Existing PPG research tends to focus on heartbeat only, and notch, an important biomarker of a PPG, is considered. As there is a tendency to not be able to do this, we tried to solve this problem; after learning, it was confirmed that malar and forehead areas were recognized as important parts of learning, through BBM analysis.

The rPPG task required various physiological knowledge, and it was challenging to apply it properly. From the point of view of comparative evaluation of the rPPG algorithm, a standardized evaluation method has not yet been prepared, and the dataset used in some prominent papers is inaccessible. In this study, the goal was to infer accurate PPGs in a short time using the axis projection method, and the V4V and UBFC datasets were selected as a result of the problem of not being able to access multiple datasets. In the case of V4V data, only the training data were disclosed; validated data could not be obtained, so the training data was divided and used for learning and evaluation. In future research, we will improve the performance of the proposed axis projection method, and create a general-purpose model using various data. In addition, we intend to propose a baseline for the rPPG method by comparing and evaluating several representative deep learning-based models, with the same datasets and evaluation metrics.

## Figures and Tables

**Figure 1 bioengineering-09-00638-f001:**
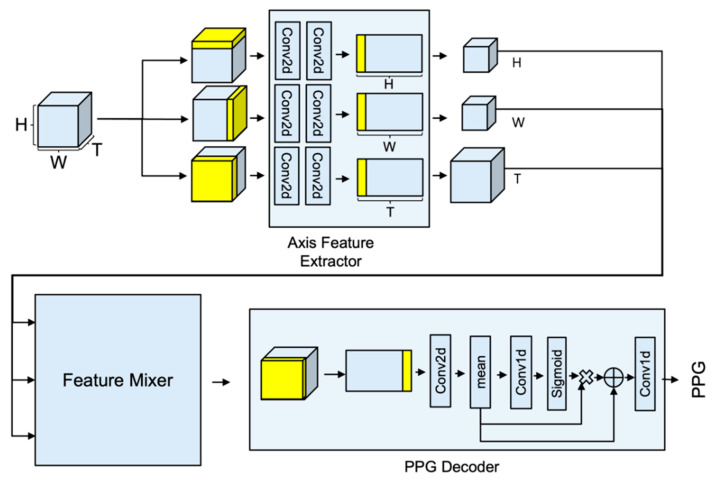
APNet architecture.

**Figure 2 bioengineering-09-00638-f002:**
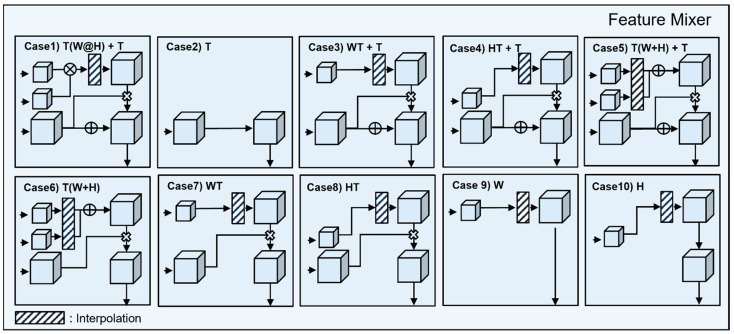
Type of feature mixer.

**Figure 3 bioengineering-09-00638-f003:**
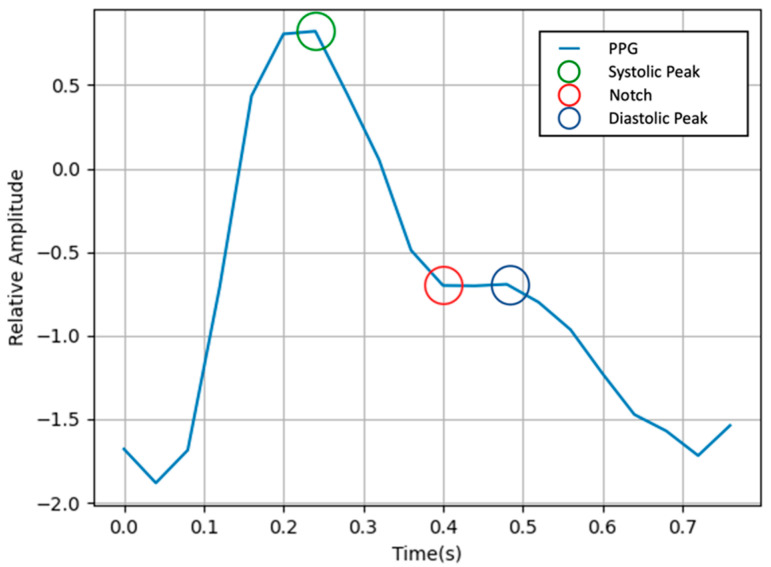
Waveform of a PPG.

**Figure 4 bioengineering-09-00638-f004:**
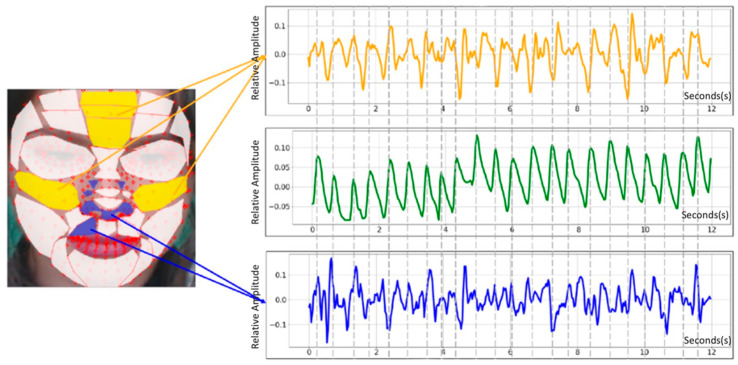
PPG extraction accuracy evaluation according to face area. (Top-5: yellow, Bot-5: blue).

**Figure 5 bioengineering-09-00638-f005:**
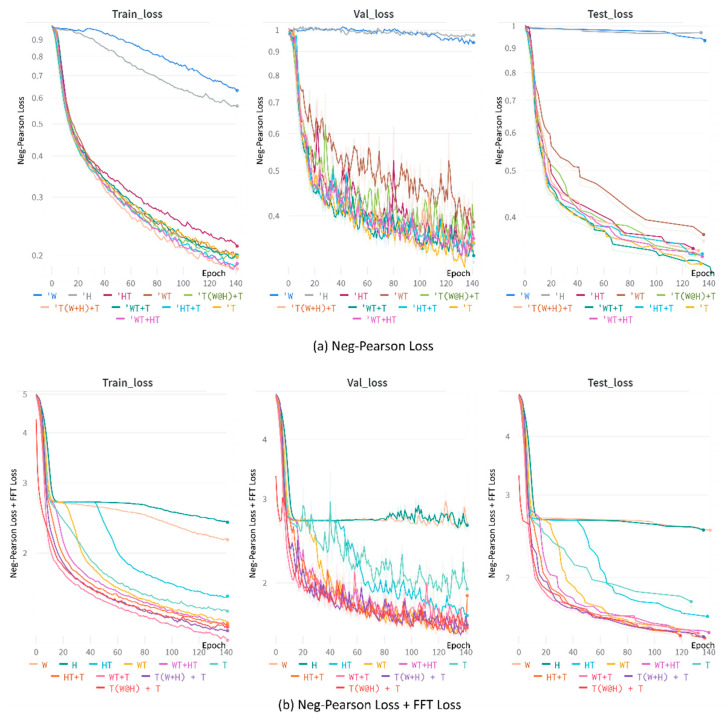
Loss graphs of APNET. (**a**) Neg-Pearson Loss; (**b**) Neg-Pearson loss + FFT Loss.

**Figure 6 bioengineering-09-00638-f006:**
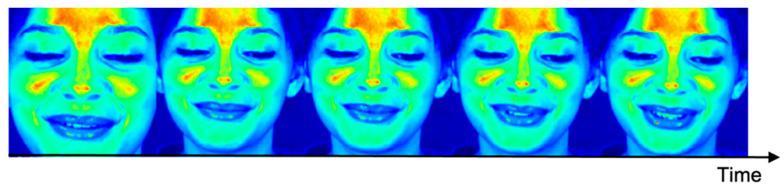
APNET LRP results.

**Figure 7 bioengineering-09-00638-f007:**
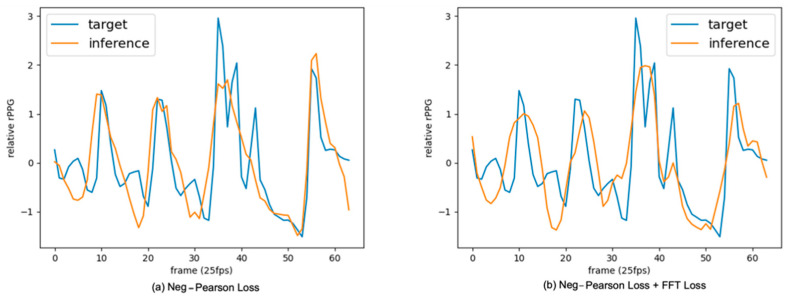
APNET’s inference results. (**a**) Neg-Pearson Loss, (**b**) neg-Pearson Loss + FFT Loss.

**Table 1 bioengineering-09-00638-t001:** Disease information according to the frequency band.

Parameter	Frequency Band	Description
ULF	≤0.003 Hz	Associated with acute heart attack and arrhythmia
VLF	0.033 Hz~0.04 Hz	Variables dependent on the renin–angiotensin system
LF	0.04 Hz~0.15 Hz	Controlled by the sympathetic and parasympathetic nervous systems
HF	0.15 Hz~0.4 Hz	There is heart rate variability related to the respiratory system, called respiratory arrhythmia

**Table 2 bioengineering-09-00638-t002:** Early warning score [3].

Score	0	1	2	3
Heart rate (beats per minute)	51–100	40–50 101–110	<40 111–130	>130
Systolic blood pressure (mmHg)	10 –160	81–100 161–200	70–80 >200	<70
Respiratory rate (per minute)	9–14	15–20	<9 21–30	>30
Temperature (degrees Celsius)	36.1–37.5	35.0–36.0 >37.5	<35	
Consciousness level	Alert	Responds to voice	Responds to pain	Unresponsive

**Table 3 bioengineering-09-00638-t003:** Ten types of feature mixer.

Feature Mixer Case #	Equation	Feature Mixer Case #	Equation
1	$Interp\left(E^{W}\cdot E^{H}\right)⨀E^{T}+E^{T}$	6	$\left(Interp\left(E^{W}\right)+Interp\left(E^{H}\right)\right)⨀E^{T}$
2	$E^{T}$	7	$Interp\left(E^{W}\right)⨀E^{T}$
3	$Interp\left(E^{W}\right)⨀E^{T}+E^{T}$	8	$Interp\left(E^{H}\right)⨀E^{T}$
4	$Interp\left(E^{H}\right)⨀E^{T}+E^{T}$	9	$Interp\left(E^{W}\right)$
5	$\left(Interp\left(E^{W}\right)+Interp\left(E^{H}\right)\right)⨀E^{T}+E^{T}$	10	$Interp\left(E^{H}\right)$

**Table 4 bioengineering-09-00638-t004:** APNET model architecture.

APNET Configuration					
Input Shape			(B, C, T: 32, H: 128, W: 128)		
Axis Extractor Module Configure					
*T*-Axis		*H*-Axis		*W*-Axis	
Transpose	(B, T, C, H, W)	Transpose	(B, H, C, T, W)	Transpose	(B, W, C, T, H)
Reshape	(B × T, C, H, W)	Reshape	(B × H, C, T, W)	Reshape	(B × W, C, T, H)
Conv2D	(B × T, C, H/2, W/2)	Conv2D	(B × H, C, T/2, W/2)	Conv2D	(B × W, C, T/2, H/2)
Conv2D	(B × T, C, H/4, W/4)	Conv2D	(B × H, C, T/4, W/4)	Conv2D	(B × W, C, T/4, H/4)
Reshape	(B, T, C, H/4, W/4)	Reshape	(B, H, C, T/4, W/4)	Reshape	(B, W, C, T/4, H/4)
Transpose	(B, C, T, H/4, W/4)	Transpose	(B, C, H, T/4, W/4)	Transpose	(B, C, W, T/4, H/4)
Reshape	(B, C, T, H/4 × W/4)	Reshape	(B, C, H, T/4 × W/4)	Reshape	(B, C, W, T/4 × H/4)
MaxViT	(B, C, T, H/4 × W/4)	MaxVit	(B, C, H, T/4 × W/4)	MaxVit	(B, C, W, T/4 × H/4)
SPAtt *	(B, C, T, H/4 × W/4)	SPAtt *	(B, C, H, T/4 × W/4)	SPAtt *	(B, C, W, T/4 × H/4)
Adaptive pool	(B, C, T, 16)	Reshape	(B, C, H, 4, 4)	Reshape	(B, C, W, 4, 4)
Reshape	(B, C, T, 4, 4)	Transpose	(B, C, 4, H, 4)	Transpose	(B, C, 4, W, 4)
Feature Mixer			(B, C, T, 4, 4)		
PPG Decoder					
Reshape			(B, C, T, 4 × 4)		
Conv2D			(B, T, T, 4 × 4)		
Mean			(B, T, T)		
Residual block			(B, T, T)		
Conv1D			(B,1, T)		
Output shape			(B, T)		

* SPAtt: Spatial Attention Module.

**Table 5 bioengineering-09-00638-t005:** Evaluation results for 10 feature mixers using the V4V dataset.

Feature Mixer		HR-MAE	HR-RMSE	R	HR-MAE	HR-RMSE	R
		Neg-Pearson Loss			Neg-Pearson Loss + FFT Loss		
1	T(W@H) + T	5.48	7.86	0.776	8.33	12.60	0.746
2	T(W + H) + T	6.18	9.62	0.666	8.05	12.58	0.745
3	WT + T	10.25	16.13	0.600	6.13	9.23	0.75
4	HT + T	9.10	14.71	0.670	6.66	10.57	0.742
5	T	8.74	13.84	0.694	6.20	9.22	0.691
6	WT + HT	6.77	10.67	0.672	4.89	7.86	0.738
7	WT	6.39	9.72	0.763	7.06	10.01	0.720
8	HT	5.90	9.18	0.767	7.16	11.00	0.663
9	H	11.74	15.63	0.246	14.57	20.72	0.135
10	W	11.41	15.47	0.305	11.02	15.34	0.291

**Table 6 bioengineering-09-00638-t006:** Evaluation results for 10 feature mixers using the UBFC dataset.

Feature Mixer		HR-MAE	HR-RMSE	R	HR-MAE	HR-RMSE	R
		Neg-Pearson Loss			Neg-Pearson Loss + FFT Loss		
1	T(W@H) + T	1.50	1.97	0.968	1.00	1.48	0.969
2	T(W + H) + T	0.75	1.13	0.972	1.62	2.54	0.970
3	WT + T	1.80	3.03	0.972	0.53	0.77	0.970
4	HT + T	1.95	2.48	0.975	1.66	2.58	0.966
5	T	1.24	1.58	0.971	1.32	1.88	0.968
6	WT + HT	1.72	2.17	0.968	1.42	2.18	0.970
7	WT	1.42	1.91	0.971	1.85	2.56	0.970
8	HT	2.32	3.53	0.970	1.53	2.19	0.970
9	H	11.66	14.75	0.716	9.85	13.79	0.770
10	W	10.35	14.85	0.796	13.05	17.46	0.739

**Table 7 bioengineering-09-00638-t007:** Comparison of quantitative assessment results with representative algorithms.

	V4V			UBFC		
Model	HR-MAE	HR-RMSE	R	HR-MAE	HR-RMSE	R
Deepphys	10.2	13.25	0.454	5.97	7.42	0.531
PhysNet	13.15	19.23	0.746	4.54	7.65	0.931
APNET(Proposed)	4.89	7.86	0.738	0.53	0.77	0.970

**Table 8 bioengineering-09-00638-t008:** Comparison of memory size with representative algorithms.

Model	Memory Size
Deepphys	5.6 MB
PhysNet	3.0 MB
APNET (Proposed)	1.1 MB

## Data Availability

The V4V dataset is available at https://vision4vitals.github.io (accessed date 19 September 2022). The UBFC dataset is available at https://sites.google.com/view/ybenezeth/ubfcrppg (accessed date 22 October 2022).

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
