# Peer review of "A Study of Projection-Based Attentive Spatial–Temporal Map for Remote Photoplethysmography Measurement"

_bioengineering, 2022, doi:10.3390/bioengineering9110638_

Round 1

Reviewer 1 Report

The manuscript proposes the Axis Projection Network (AP-Net), which complements the point that the 3D CNN method requires a large memory, the STMap method requires a preprocessing method, and the dyschromatic reflection model (DRM) method does not learn long-term temporal characteristics. 

The proposed method is interesting, but some demonstration and experimental study should be stated clearly:

1. More details (e.g., mathematical formulations ) about the Feature Mixer in Section 3.2 and Figure 2 should be given.

2. Besides the study of the proposed AP-Net, some state-of-the-art results in V4V should be given for comparison. For example, in Table 4, besides the memory size, the performance of DeepPhys, PhysNet, and APNet should be given.

3.  It is good to show several background surveys about recent rPPG measurement methods for broader readers. Please consider introducing some related works such as [A][B]:

[A] Video-Based Heart Rate Measurement Recent and Future Prospects, IEEE TIM 2018

[B]  Facial-Video-Based Physiological Signal Measurement: Recent advances and affective applications, IEEE SPM 2021

Author Response

Dear Reviewer,

Thanks for interesting in my research.

Additional experiments were conducted during major revision, and the structure of the PPG Decoder among the structures of the proposed model was partially changed. By removing the MaxVit layer from the PPG Decoder and changing it to Conv2D, it is stable even in Neg Pearson Loss.

  1.  More details (e.g., mathematical formulations ) about the Feature Mixer in Section 3.2 and Figure 2 should be given.
    A:  Add Feature Mixer formula in Table 3. And then add more details Table 4
  2.  Besides the study of the proposed AP-Net, some state-of-the-art results in V4V should be given for comparison. For example, in Table 4, besides the memory size, the performance of DeepPhys, PhysNet, and APNet should be given. 
    A: add the Table 6. Table 6 show the quantitative assessment of representative rPPG algorithms.

Reviewer 2 Report

I think the work of this paper is relatively comprehensive. The most obvious problem is that all the line figures are not professional enough.

Fig. 3, the coordinates have no units.

Fig. 4, there's no coordinates headings, the numbers on the coordinates are too small to see clearly.

There are two Fig. 4? It should be Fig. 5. Please make the figure on the common scientific drawing software, and indicate the horizontal and vertical coordinates. At the same time, it should have a good definition.

Fig. 5? It should be Fig. 6. Please use a, b and c to distinguish the different pictures and write the differences in the caption of the figure.

In addition, please cite the latest relevant research in this journal and make a brief comparative analysis.

If authors can draw (summarize) the main innovation points of this paper, it will make the paper more readable.

Author Response

Dear Reviewer,

Thanks for interesting in my research.

Additional experiments were conducted during major revision, and the structure of the PPG Decoder among the structures of the proposed model was partially changed. By removing the MaxVit layer from the PPG Decoder and changing it to Conv2D, it is stable even in Neg Pearson Loss.

I checked all of the figure number and Table number. 

Dear Reviewer,

Thanks for interesting in my research.

Additional experiments were conducted during major revision, and the structure of the PPG Decoder among the structures of the proposed model was partially changed. By removing the MaxVit layer from the PPG Decoder and changing it to Conv2D, it is stable even in Neg Pearson Loss.

I checked all of the figure number and Table number. And then add the Table 3,4,6 for make readable.

Round 2

Reviewer 1 Report

It looks good in the current version.

One suggestion is to add two classical overview literature in the introduction part to benefit broader readers.

[A] Facial-Video-Based Physiological Signal Measurement: Recent advances and affective applications, IEEE SPM 2022

[B] Video-Based Heart Rate Measurement: Recent Advances and Future Prospects, IEEE TIM 2020

Author Response

Dear Reviewer.

The content order and reference format have been modified to match the "Instructions for Authors".
In addition, unnecessary English words have been deleted from the Abstract.
An additional experiment was conducted on one dataset, and finally, the quality of English was improved overall.

Reviewer 2 Report

Accepted.

Thanks for the authors response. All my concerns are set now.

Author Response

(The authors gave the same response as above.)
